



# On Quadruplet Interactions for Ocean Surface Waves

Adhi Susilo[1], Will Perrie[1], and Bash Toulany[1]

[1]Fisheries and Oceans Canada, Bedford Institute of Oceanography, Dartmouth, Nova Scotia, Canada

*Correspondence to:* A Susilo (asusilo@dal.ca)

**Abstract.** Nonlinear wave-wave interactions among ocean surface waves are dominated by quadruplet wave-wave interactions. Computing the nonlinear 4-wave interactions with the Boltzmann integral requires finding the loci of interactions for the quadruplets or solving the delta functions. This is an important part of the computation, but so far it is done by an iteration method that consumes computational time and may not converge after doing several iterations. In this paper, an explicit methodology to find the loci of the quadruplet interactions is presented. This research target is to develop a better method to compute the loci. To illustrate the method, there are 4 cases that will be discussed in this paper. Results show that the new method gives better results than the previous methods that have been applied. Moreover, without iterations the presented method requires less computational loops and some variables, for example the distance between loci, denoted $ds$, can be determined without any looping. Therefore, the new method leads to better and faster computations than the previous iteration method.

## 1 Introduction

In modeling wind generated waves, there are three major factors that must be taken account, which are wind input, dissipation, and non linear wave-wave interactions. Mathematically, these sources are written as, respectively, $S_{IN}$, $S_{DS}$, and $S_{NL}$. See The SWAMP Group (1985), Tolman (2009), Perrie et al. (2013), and Tolman et al. (2014) for details. The first term gives the energy to the waves from the wind, the second term reduces the energy due to wave breaking dissipation, and the last term is conservative, transferring energy among the wavenumbers of the spectrum, Komen et al. (1994), Cavaleri et al. (2007), and Holtuijsen (2007). In general, nonlinear wave-wave interactions ($S_{NL}$) among the ocean surface waves are dominated by quadruplet wave-wave interactions, but in shallow water, triad wave-wave interactions may also exist.

Computing the nonlinear 4-wave interactions can be done by so-called exact methods (using the Boltzmann integral) as described by Tracy & Resio (1982), Resio & Perrie (1991), and Van Vledder (2006). Alternately, for operational forecasts we can use approximation methods, such as DIA, the *discrete interaction approximation* (WAMDIG (1988)). In practical operational marine forecasts, wave models usually use the DIA method, Tolman (2009), but the exact method is also important as a benchmark to solve the problem when the approximation method fails to accurately predict the wave spectra due to unusual wind inputs (for example: rapidly developing storms or hurricanes).

In this paper, we discuss the exact method, partially solving the non-linear interactions with the Boltzmann integral. However, we do not derive all equations of the integral, but focus on a small part of the integration, the important part, the starting point, the domain of the integration.





We begin with the main equation, the Boltzmann integral, as shown on Eq. 1. Thereafter, the methodology to solve the equation is explained. Moreover, solving the integral is beyond the scope of this paper. That is done in many other places, for example Resio & Perrie (1991). However, for any definite integral, we have to integrate it over its domain. Regarding the computation, the Boltzmann equation integral is approximated by a collection of nested do-loops; the smaller step the

increment in wavenumber space, $dk$, the better is the result. But on the other hand, when $dk$ is quite small, the computation time becomes excessively large. However, the computation time can be reduced if we do not compute the unnecessary terms (e.g. terms that are zero, or close to zero). The objective of this study is to determine the domain of the integration only, which is the basis to efficiently solve the Boltzmann integral. The content of this paper is to explain the new method mathematically and physically.

As defined by Webb (1978), the Boltzmann integral of the nonlinear action density transfer for quadruplet nonlinear interactions may be written as:

$$\frac{\partial n_1}{\partial t} = \iiint C(\mathbf{k}_1, \mathbf{k}_2, \mathbf{k}_3, \mathbf{k}_4)\, \delta(\mathbf{k}_1 + \mathbf{k}_2 - \mathbf{k}_3 - \mathbf{k}_4)\, \delta(\omega_1 + \omega_2 - \omega_3 - \omega_4)$$

$$[n_1 n_3 (n_2 - n_4) + n_2 n_4 (n_3 - n_1)]\, d\mathbf{k}_2 d\mathbf{k}_3 d\mathbf{k}_4, \tag{1}$$

where $\mathbf{k}_i$ are vectors. Thus it is a six-fold integral in terms of $k_x$ and $k_y$. It is too costly to use in operational wave forecasting

and therefore this term must be approximated if it is to be implemented in practical applications (see Perrie et al. (2013)).

As you see, the integration contains delta functions $\delta(X)$. The property of this function says that the integration of $\delta(X)$ is zero if $X \neq 0$ and one if $X = 0$, so we only compute Eq. (1), on the domain where the following conditions are satisfied

$$\mathbf{k}_1 + \mathbf{k}_2 - \mathbf{k}_3 - \mathbf{k}_4 = 0 \tag{2}$$

$$\omega_1 + \omega_2 - \omega_3 - \omega_4 = 0. \tag{3}$$

Therefore, determining Eq. (2) and Eq. (3) is the important first part of the process to compute the Boltzmann integral. Points that satisfy those conditions are *loci* of the quadruplet wave-wave interactions. This is the main topic of this research, finding the loci or domain of the integration. The new method is expected to give better performance than the previous method.

In wave models, usually $\mathbf{k}_1$ and $\mathbf{k}_3$ are given as *incident wave numbers*, and then we find $\mathbf{k}_2$ and $\mathbf{k}_4$ using the dispersion relation

$$\omega^2 = gk \tanh(kd) \tag{4}$$

where $g$ is the gravitational acceleration, $d$ is water depth.

In some circumstances, finding $\mathbf{k}_2$ and $\mathbf{k}_4$ is not straightforward. Existing computational methods use iteration schemes, something like

$$X_{new} = X_{old} + \epsilon, \tag{5}$$

where $\epsilon$ is some error, and the computation is stopped when $\epsilon \approx 0$. This method can take excessive time to solve the dispersion relation Eq. (4) and sometimes the iteration does not converge. The computational time can be reduced if we can solve this problem without iterating.





Tracy & Resio (1982) shows how to reduce the computational time by using a specific grid geometry. Their work shows that the integrand of Eq. (1) is the product of functions: *F(action density)* and *G(geometry term)*. In polar coordinates, the grid geometry of Tracy & Resio (1982) allows the spectrum to be expressed in terms of wavenumbers $k_i$ such that $k_i = \lambda k_{i-1}$. Thus, it can be shown that,

$$G_i = \lambda^{15/2} \times G_{i-1}. \tag{6}$$

The computation time is diminished, because the iteration is done for the basic geometry ($G_0$) only. However, the spectral grid is not flexible, because it is a function of the multiplication factor $\lambda$. In some applications, it may be desirable to design the grid so that the spectral peak is always exactly at a specific wavenumber, for example the 15th wavenumber ($k_{15}$), and there are always 15 spectral bins to represent the spectral forward face ($k \leq k_{15}$), as the spectrum evolves in time (Resio &

Perrie (1991)). However to simulate this case, the basic geometry ($G_0$) needs to be computed at each time step, using the grid geometry of Tracy & Resio (1982) because the spectral peak downshifts to lower frequencies with time. Clearly this methodology is very expensive.

## 2   The explicit methods

This is the main section of this paper. Finding loci of the interactions are explained thoroughly. We are to going to solve Eq.

(2) and Eq. (3) without iteration to illustrate the method. There are 4 cases that will be discussed. These cases are functions of vector $\mathbf{p}$, angular velocity $q$, and a dimensionless variable $kd$.

The defined variables are $\mathbf{p}$ and $q$, where $\mathbf{p}$ is a vector, stated as

$$\mathbf{p} = \mathbf{k}_3 - \mathbf{k}_1 = \mathbf{k}_2 - \mathbf{k}_4, \tag{7}$$

and its components may be written as,

$$p_x = k_{3x} - k_{1x} = k_{2x} - k_{4x} \tag{8}$$
$$p_y = k_{3y} - k_{1y} = k_{2y} - k_{4y}, \tag{9}$$

and the magnitude is

$$p = \sqrt{p_x^2 + p_y^2}. \tag{10}$$

The $q$ variable is an angular velocity difference

$$q = \omega_3 - \omega_1 = \omega_2 - \omega_4. \tag{11}$$

As you see, the vector $\mathbf{p}$ is a function of $\mathbf{k}_1$ and $\mathbf{k}_3$, so the four cases we are going to consider are determined by vectors $\mathbf{k}_1$ and $\mathbf{k}_3$:





1. First case, where vector $\mathbf{k}_1$ equals to vector $\mathbf{k}_3$.

2. Second case, where the magnitude of vector $\mathbf{k}_1$ equals to the magnitude of vector $\mathbf{k}_3$.

3. Third case, where vector $\mathbf{k}_1$ does not equal to vector $\mathbf{k}_3$ and in deep water.

4. Fourth case, where vector $\mathbf{k}_1$ does not equal to vector $\mathbf{k}_3$ and in shallow water.

Following subsections will show each case thoroughly. Readers may skip the derived equations, they can find the main idea of each case which is summarized at the end of each subsection in italics. Please keep in mind that the derived equations for Case III or IV are built by defining that $\mathbf{p}$ is lying on the x-axis as summarized in Section 3.

## 2.1   Case I, $\mathbf{k}_1 = \mathbf{k}_3$

In this case, the $p$ and $q$ are zero, and $\mathbf{k}_2 = \mathbf{k}_4$. This case will give infinite pairs of $\mathbf{k}_2$ and $\mathbf{k}_4$, but the term $[n_1 n_3(n_4 - n_2) +$
$n_2 n_4(n_3 - n_1)]$ will be zero, because $n_4 = n_2$ and $n_3 = n_1$. Therefore, we *do not need to find any loci* of $\mathbf{k}_2$ and $\mathbf{k}_4$ for this case.

## 2.2   Case II, $|\mathbf{k}_1| = |\mathbf{k}_3|$

The variable $q$ will be zero but $p \neq 0$. For this case, the loci are not a function of depth, $d$. We observe that the loci of $\mathbf{k}_2$ and $\mathbf{k}_4$ are normal lines to the slope of $\mathbf{p}$, and some methods bound the normal lines by a circle with radii $k_{max} = f(4\omega_1)$, to limit
the computation.

The slope of $\mathbf{p}$ is stated as,

$$m_p = \frac{p_y}{p_x} \tag{12}$$

then the slope normal to the $\mathbf{p}$ will be

$$m_n = -\frac{1}{m_p} = -\frac{p_x}{p_y}. \tag{13}$$

Therefore, the normal line is defined as,

$$y = m_n(x - x_0) + y_0. \tag{14}$$

For this case $(x_0, y_0) = (k_{3x}, k_{3y})$, and so

$$y = m_n(x - k_{3x}) + k_{3y}. \tag{15}$$

As you see, *the loci of this case are lines that are perpendicular to vector* $\mathbf{p}$.

In the next section, the loci produce an egg-like shape, as shown on Figure 1. In this paper, the terms 'circle', 'ellipse', and 'egg-like' are interchangeably used to describe the loci of Case III and IV. It is called a circle because it has a center and radii and a starting shape to get the loci. It is called an ellipse because it has a major axis and this shape mimics the real loci. Finally, it is called an egg-like shape because the loci shape really is like an egg.





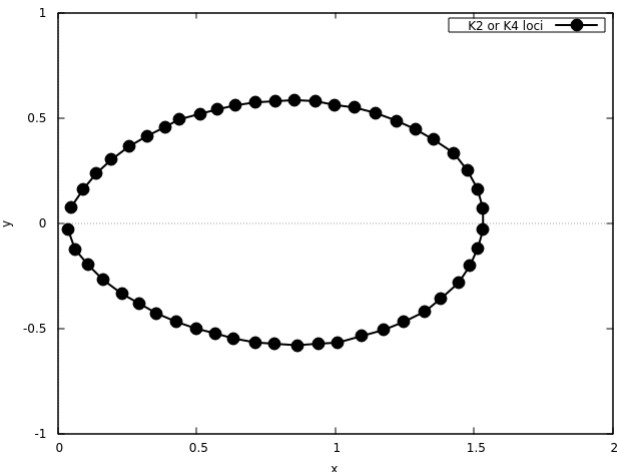

**Figure 1.** An example of loci from Case III or IV.

## 2.3 Case III, $k_1 \neq k_3$, in deep water

The loci of $\mathbf{k}_2$ and $\mathbf{k}_4$ are on an egg-like shape with its major axis parallel to the vector $\mathbf{p}$. Therefore, we have to find points on the 'end points' of the maximum secant line to get its diameter. These 'end points' are called $r_{min}$ and $r_{max}$ respectively.

### 2.3.1 Finding $r_{min}$

In the dispersion relation Eq. (4), we have $\tanh(kd) \approx 1$ when $kd \geq 3$, corresponding to the deep water approximation. Therefore, we rewrite Eq. (4) in a simpler way

$$\omega^2 = gk. \tag{16}$$

From the definition of $q$ in Eq. (11) and from Eq. (16), we have

$$q = \sqrt{g}(\sqrt{k_2} - \sqrt{k_4}). \tag{17}$$

From the definition of $\mathbf{p}$ in Eq. (7) and taking its magnitude and the magnitudes of the $\mathbf{k}_i$ vectors, and remember that these vectors are lying on the x-axis, we can get

$$p = k_3 - k_1 = k_2 - k_4. \tag{18}$$

Therefore, we can write,

$$k_4 = k_2 - p. \tag{19}$$

Let $k_2 = r_{min}p$, so Eq. (19) becomes

$$k_4 = r_{min}p - p = (r_{min} - 1)p. \tag{20}$$



Equation Eq. (20) gives negative values for $k_4$ when the value for $r_{min}$ is less than 1. However, we need only be concerned about the magnitude of $\mathbf{k}_4$. Thus we redefine $k_4$ as

$$k_4 = (1 - r_{min})p. \tag{21}$$

We substitute $k_2 = r_{min}p$ and Eq. (21) into Eq. (17), which gives

$$q = \sqrt{g}\left[\sqrt{r_{min}p} - \sqrt{(1 - r_{min})p}\right], \tag{22}$$

we then take the $\sqrt{p}$ out of the bracket and do some algebraic re-arrangement, which gives

$$\left[0.5 - \frac{q^2}{2gp}\right]^2 = r_{min} - r_{min}^2. \tag{23}$$

Therefore, we get a quadratic function, $r_{min}^2 - r_{min} + C = 0$, where

$$C = \left[0.5 - \frac{q^2}{2gp}\right]^2. \tag{24}$$

We solve for the roots of Eq. (23)

$$r_{min_{1,2}} = \frac{1 \pm \sqrt{1 - 4C}}{2}. \tag{25}$$

For the computation, we define $r_{min} \geq 0.5$, which implies that we should therefore use the bigger root, which is

$$r_{min} = \frac{1 + \sqrt{1 - 4C}}{2}. \tag{26}$$

### 2.3.2 Finding $r_{max}$

For $k_2 = r_{max}p$, we can write,

$$k_4 = r_{max}p - p = (r_{max} - 1)p. \tag{27}$$

We substitute $k_2 = r_{max}p$ and Eq. (27) into Eq. (17) which gives

$$q = \sqrt{g}\left[\sqrt{r_{max}p} - \sqrt{(r_{max} - 1)p}\right], \tag{28}$$

and with some algebraic re-arrangement, we find

$$\sqrt{r_{max}(r_{max} - 1)} = r_{max} - \left[\frac{q^2}{2gp} + 0.5\right], \tag{29}$$

and we simplify Eq. (29) as $\sqrt{r_{max}^2 - r_{max}} = r_{max} - C$, where

$$C = \left[\frac{q^2}{2gp} + 0.5\right]. \tag{30}$$

Then squaring both sides of Eq. (29), we get

$$r_{max} = \frac{C^2}{(2C - 1)}. \tag{31}$$





### 2.3.3 Finding the loci

After $r_{min}$ and $r_{max}$ are found, now we assume that the loci of $\mathbf{k}_2$ lie on a circle-like shape with a center and radius, $c_c$ and $c_r$, respectively

$$c_c = \frac{(r_{max} + r_{min})p}{2} \tag{32}$$

$$c_r = \frac{(r_{max} - r_{min})p}{2}. \tag{33}$$

If we slice the circle-like structure of the loci into $n$ parts, and if the angle of each slice is $d\phi$, then using the triangle identity, we compute $k_2$ as

$$k_2^2 = c_c^2 + c_r^2 - 2c_c c_r \cos(d\phi). \tag{34}$$

Therefore, we know the magnitude of $\mathbf{k}_2$. To find its direction, we must use Eq. (17), and rewrite it as, $q = \sqrt{g}(\sqrt{k_2} - \sqrt{k_2 - p})$. Therefore, we will get

$$k_2 - p = \zeta \tag{35}$$

where $\zeta = \left[\sqrt{k_2} - \frac{q}{\sqrt{g}}\right]^2$. We continue with the algebraic derivation by defining the $p$ value which is lying on our $x$-axis, $(p, 0)$. Therefore, we obtain

$$(k_{2x} - p)^2 + k_{2y}^2 = \zeta^2. \tag{36}$$

The above equation specifies another circle with the center at $p$ and radius $\zeta$. We continue with the solution of Eq. (36) and substitute $k_2$ in polar coordinate form; $(k_2 \cos\theta, k_2 \sin\theta)$. Therefore, we get

$$k_2^2 - 2pk_2 \cos\theta + p^2 = \zeta^2. \tag{37}$$

Therefore, the angle between $k_2$ and our axis is defined as

$$\cos\theta = \frac{k_2^2 + p^2 - \zeta^2}{2pk_2}. \tag{38}$$

Finally, using Eq. (34) and Eq. (38), we can determine the magnitude and direction of $\mathbf{k}_2$, and once $\mathbf{k}_2$ is determined, the $\mathbf{k}_4$ is found. Loci of the tips of these vectors depict an egg-like shape.

As one can see, in this case *we assume* $\tanh(kd) \approx 1$, *to simplify the problem. We compute the minimum point (tip of the smallest vector) and maximum point (tip of the largest vector). Next we find the center of this interval, then other points (tip of vectors in between) surrounding the center can be determined.* Figure 2 depicts this process.





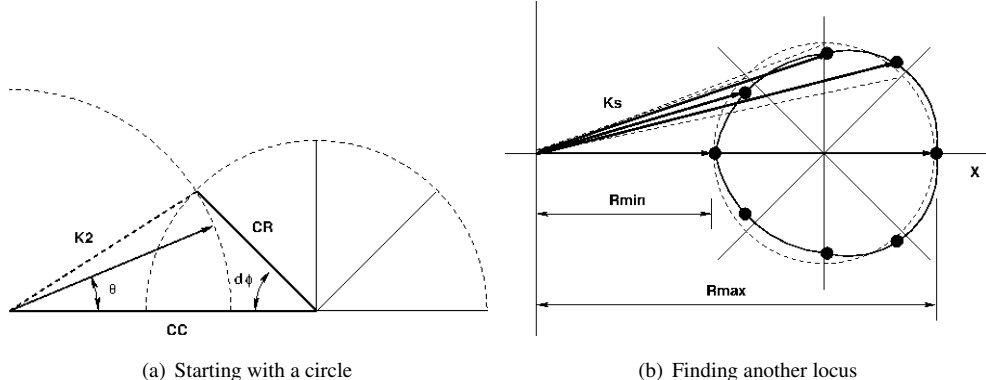

|  (a) Starting with a circle | (b) Finding another locus |

**Figure 2.** After the minimum and maximum vector are found, other vectors in between are determined.

## 2.4 Case IV, $k_1 \neq k_3$, in shallow water

As in Case III, the loci of $k_2$ and $k_4$ are also an egg-like shape with its center parallel to the vector **p**. However, the points on the 'ends' of the maximum secant line are found by a linear approximation method, in this case.

### 2.4.1 Finding $r_{min}$

Because the $kd$ is not large compared to 1, therefore we *cannot* neglect the term, $\tanh(kd)$. Thus, from Eq. (11) and Eq. (4) and substituting values, $k_2 = r_{min}p$ and $k_4 = (1 - r_{min})p$ respectively, we get

$$q = \sqrt{g[r_{min}p]\tanh([r_{min}p]d)} - \sqrt{g[(1-r_{min})p]\tanh([(1-r_{min})p]d)}. \tag{39}$$

The square root is a multiple products function. To find $r_{min}$ is not a straightforward problem. Previous efforts tend to try to solve the problem without simplifying it first. Most of these approaches use an iteration method to find the unknown roots. An example is the computation done by Resio & Perrie (2008).

In this study, we use another approach, which is straightforward and without iteration. In this approach, we propose an explicit analytic solution, which can find the solution easily and much faster than the previous iterative approaches.

Let's go back to Eq. (39). Before we solve this equation, we focus on the dependent variable, $r_{min}$.

Let's call this variable $x$. Therefore, we get

$$q = \sqrt{g[x]p\tanh([x]pd)} - \sqrt{g[(1-x)]p\tanh([1-x]pd)}. \tag{40}$$

We rewrite this equation in a simpler way

$$q = f(x)g(x) - f(1-x)g(1-x) = H(x) - H(1-x) \tag{41}$$

where $H = fg$. Now, we can see clearly what we have. The $q$ variable is just a difference between 2 functions. In fact they are the same function, but the second term on the right hand side is a mirror, $f(-x)$, of the first one, $f(x)$, and also shifted +1, as $f(1-x)$.


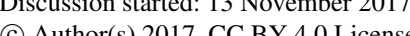


Let's examine the behaviors of $H(x)$ and $H(1-x)$ and we will plot them in the domain $0 \leq x \leq 1$. We find that both equations are linear, and therefore we can conclude that the difference is also linear. For example, if we have linear functions, $f_1(x) = m_1 x$ and $f_1(-x) = -m_1(x)$, the difference of $f_1(x) - f_1(-x)$ is a linear function too, as shown below

$$f_1(x) - f_1(-x) = m_c x \tag{42}$$

where $m_c = m_1 - (-m_1) = 2m_1$. The difference is zero, when $r_{min} = x = 0.5$, which means that the function is shifted to the right. Therefore the difference function, $q$, may be stated as

$$q = m_c(x - 0.5). \tag{43}$$

Replacing $x$ with $r_{min}$, we get

$$r_{min} = \frac{q}{m_c} + 0.5. \tag{44}$$

This method is depicted on Figure 3. We find that the difference of two functions, $q$, is linear, and so we have to find its slope and shifting factors. Once we have the equation $q = f(x)$, then the variable $x$ (or $r_{min}$) will be known, because $q$ is provided.

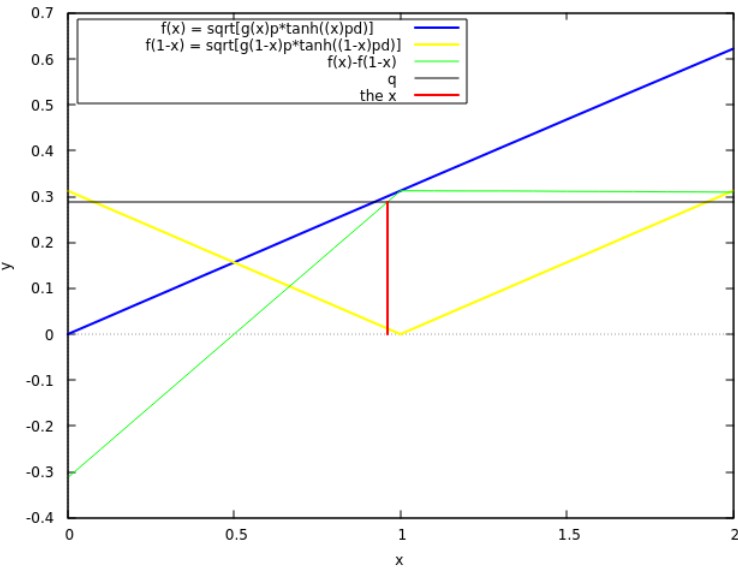

**Figure 3.** Finding the $r_{min}$. The $q = f(x) - f(1-x)$ behaves linearly on domain $x < 1$. Therefore, if $q$ is given, then $x$ or $r_{min}$ can be found.

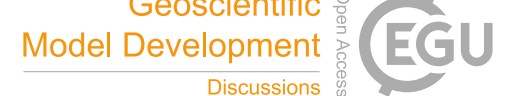



### 2.4.2 Finding $r_{max}$

Because $r_{max}$ is greater than 1, therefore we rewrite Eq. (39) as

$$q = \sqrt{g[r_{max}p]\tanh([r_{max}p]d)} - \sqrt{g[(r_{max}-1)p]\tanh([(r_{max}-1)p]d)}. \qquad (45)$$

Figure 4(a) shows that the difference, $q$, is not linear for domain $x > 1$. To find the solution, $r_{max}$, we approximate the $q$

curve by dividing the curve into several linear lines as depicted in Figure 4(b). Thus, we just focus on the segment where the line $q$ intersects with the segment line (see Figure 4(b)). This intersection is the second root or the $r_{max}$ (the solution of Eq. (45)).

For this study, the curve is divided into 10 segments, but only 6 of them are shown on Figure 4(b). On the domain $1 < x < 4$, because the rate of change of the slope of the curve is large, therefore we will use varying segments with smaller widths than

10 the segments on the domain $> 4$ to capture the intersection accurately.

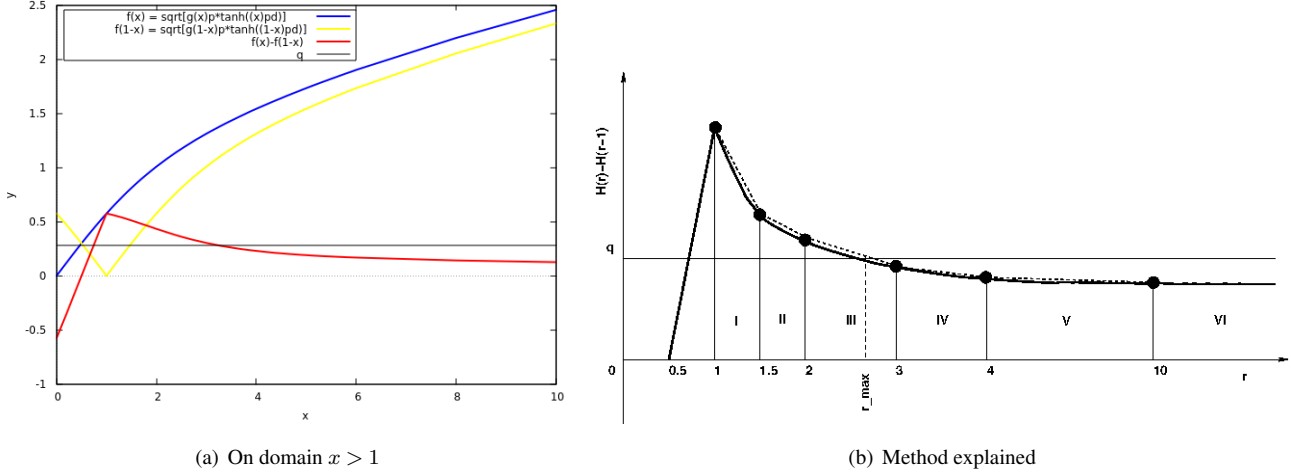

(a) On domain $x > 1$        (b) Method explained

**Figure 4.** Finding the $r_{max}$. On domain $x > 1$, the $q$ is not linear as shown on graph (a). Graph (b) shows the curve $q$ is divided into several segments. If $q$ is given, then $x$ or $r_{max}$ can be solved by finding the intersection.

### 2.4.3 Finding the loci

The same procedure is applied as in Case III; the loci of $\mathbf{k}_2$ is assumed to be a circle. Therefore, using Eq. (34), we get the magnitude of $\mathbf{k}_2$. Finding its direction, we rewrite Eq. (17) as

$$q = \sqrt{g}(\sqrt{k_2\tanh(k_2d)} - \sqrt{k_4\tanh(k_4d)}). \qquad (46)$$

Substituting and performing some algebraic re-arrangement on this equation, we get



$$k_2 - p = \frac{\left[ \sqrt{k_2 \tanh(k_2 d)} - \frac{q}{\sqrt{g}} \right]^2}{\tanh[(k_2 - p)d]}. \tag{47}$$

For this case, we cannot use Eq. (38) immediately, because Eq. (47) is an implicit equation. Before finding the $\cos\theta$, we have to first solve Eq. (47), rewrite and multiply both sides by $d$, which leads to,

$$[(k_2 - p)d] \tanh[(k_2 - p)d] = \left[ \sqrt{k_2 \tanh(k_2 d)} - \frac{q}{\sqrt{g}} \right]^2 d. \tag{48}$$

In a simpler form, this equation can be summarized as

$$X \tanh X = Y. \tag{49}$$

We use the Lambert's Continued Fraction, Weisstein (MathWorld), to solve the $\tanh X$ as,

$$\tanh X = \frac{X}{1 + \frac{X^2}{3 + \frac{X^2}{5 + \frac{X^2}{7 + \dots}}}}. \tag{50}$$

After the $X$ is found, the $\cos\theta$ can be computed, as

$$(k_2 - p)d = X. \tag{51}$$

Again, by defining the $p$ value which is lying on our $x$-axis, $(p, 0)$, and substituting $k_2$ in polar coordinate form, we get

$$\sqrt{(k_{2x} - p)^2 + k_{2y}^2} = \frac{X}{d} \tag{52}$$

$$(k_2 \cos\theta - p)^2 + (k_2 \sin\theta)^2 = \left[ \frac{X}{d} \right]^2. \tag{53}$$

Finally, we obtain,

$$\cos\theta = \frac{k_2^2 + p^2 - \left[ \frac{X}{d} \right]^2}{2pk_2}. \tag{54}$$

Then, using Eq. (51) and Eq. (54), we can determine the magnitude and direction of $\mathbf{k}_2$, and also $\mathbf{k}_4$. Loci of the tips of these vectors also depict an egg-like shape.

As one can see, in this case *we can't assume $\tanh(kd) \approx 1$. This makes the problem more difficult to solve. Previous methods use iteration and the present method proposes a simpler methodology. Similar to Case III, we compute the minimum point and*

*maximum point, and find the center of this interval. Then other points surrounding the center can be determined, see Figure 2.*





# 3 The algorithm

The numerical model is developed using straightforward equations as discussed in the previous section. It requires no iteration and uses the linearizion approach for the non-linear problem. The algorithm to find the $\mathbf{k}_2$ and $\mathbf{k}_4$ can be summarized as following,

5    – Read $\mathbf{k}_1, \mathbf{k}_3$ and depth from given input.

   – Based on these inputs, the program will select the case, from the four described above.

   – For Cases III and IV, the computation is done on the x-axis first (assume $\mathbf{p}(p, 0)$), and then the solved loci are rotated by $\theta_p$ degrees anticlockwise, where $\theta_p$ is in the direction of $\mathbf{p}$. Figure 5 depicts this mechanism.

   – Finally the loci of $\mathbf{k}_2, \mathbf{k}_4$, and $ds$ are determined, where $ds$ is the distance between two consecutive points.

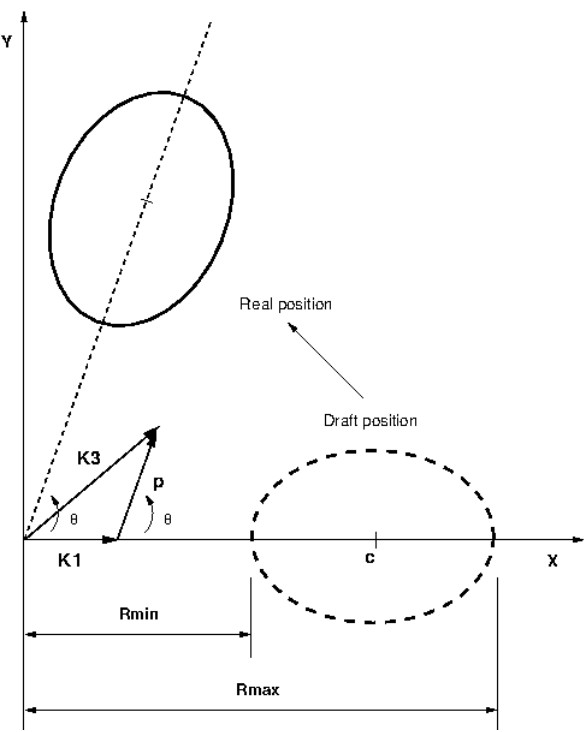

**Figure 5.** The sketch of the mechanism to get the loci.

10    The derived equations on the previous sections are based on the assumption that $\mathbf{k}_3 > \mathbf{k}_1$. If $\mathbf{k}_3 < \mathbf{k}_1$, the integral is the negative of the integration that is computed by the $\mathbf{k}_3 > \mathbf{k}_1$ condition.

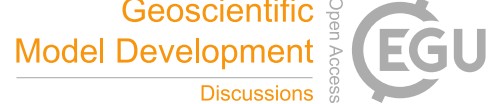



## 4    Results

The previous method by Resio & Perrie (2008) uses 50 calculations to iterate the $r_{min}$ and an error flag if the computation does not converge. To find the $r_{max}$, it requires 200 calculations for a coarse approximation and 10 additional calculations for refining it, and it is also equipped with an error flag. However, the new method requires no iteration and converges. Thus the
computational time is faster than that of the old one.

The results shown here are for the loci of $\mathbf{k}_2$ and the resonance condition, $\mathbf{k}_1 + \mathbf{k}_2 - \mathbf{k}_3 - \mathbf{k}_4 = 0$. Figures 6 through 7 show another disadvantage from the iteration method, besides being slow. Figure 7 shows results of both methods on the same axis for comparison. In the Case II, where $p \neq 0$ and $q = 0$, we should get a normal line to the vector $\mathbf{p}$. Although the difference is small, the old iterative method cannot produce a straight line as expected, especially when $\mathbf{p}$ is horizontal or vertical. See
Figures 7a and 7b for details.

For other cases, the iteration method gives a small residue (about $O(10^{-8})$) for the sum of $\mathbf{k}_1 + \mathbf{k}_2 - \mathbf{k}_3 - \mathbf{k}_4$ in Eq. (2), but the new method can achieve this condition more accurately, as shown in Figures 8a and 8b, essentially zero.

Now, we discuss the result of the requirement that the frequencies sum up to a total of zero (Eq. (3)). As explained in Section 2, finding that the summation of the wave numbers should equal zero involves $q$ (Eq. (11)), the angular velocity difference.
Therefore, once the loci are found, we can conclude that the total sum of angular velocities is also zero. Figure 9 illustrates the resonance for Case II, the relations between total $\mathbf{k}$s and total $\omega$. It is shown that $|\mathbf{k}_1| = |\mathbf{k}_3|$ and $|\mathbf{k}_2| = |\mathbf{k}_4|$, therefore $\omega_1 = \omega_3$ and $\omega_2 = \omega_4$ regardless the water is finite depth with $\tanh(kd)$ or not. A sample of resonance conditions for Cases III and IV are tabulated in Table 1 and Table 2 respectively.

Because the solution is straightforward and because no iteration is needed, setting up the grid is not necessary, because the
new method can work for any grid. This is clearly an improvement over the iterative method.

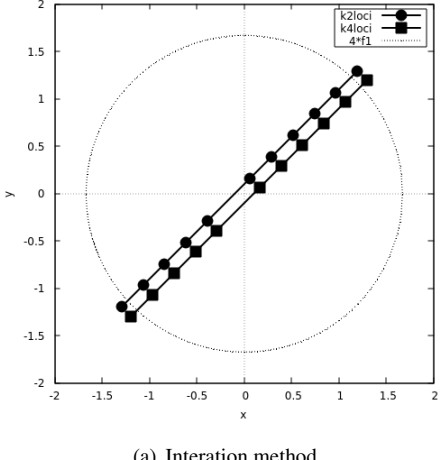

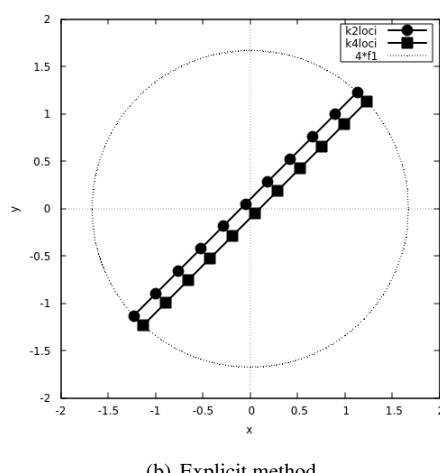

(a) Interation method                                           (b) Explicit method

**Figure 6.** Results from different methods for Case II with $\mathbf{p}$ have the some slope. The iteration method, graph (a), gives non uniform $ds$ and exceeds the boundary $4f$. On the other side, graph (b), the explicit method gives uniform $ds$ and does not exceed the boundary.





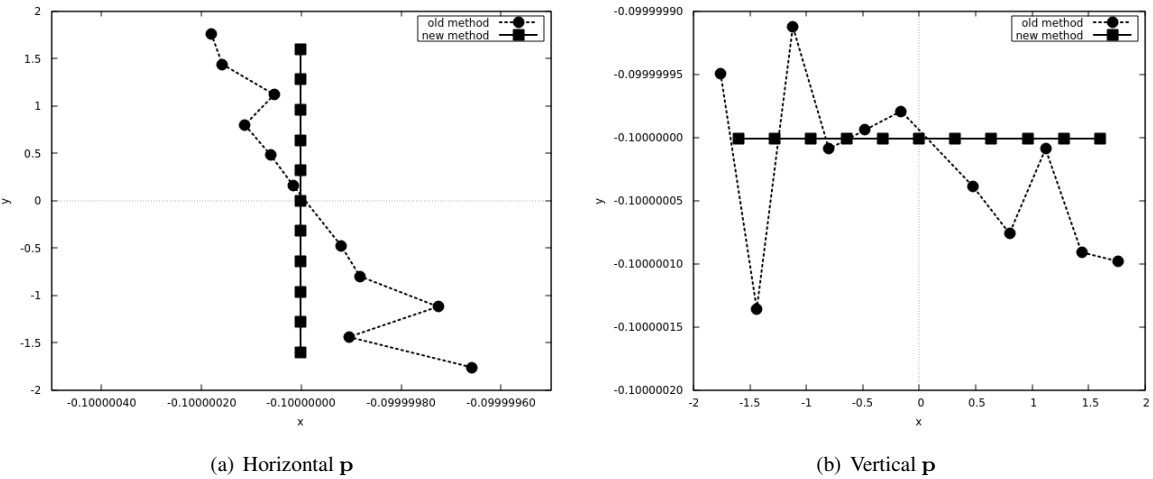

(a) Horizontal **p**    (b) Vertical **p**

**Figure 7.** As in Figure 6, but **p** is either horizontal or vertical and only $\mathbf{k}_2$ loci shown.

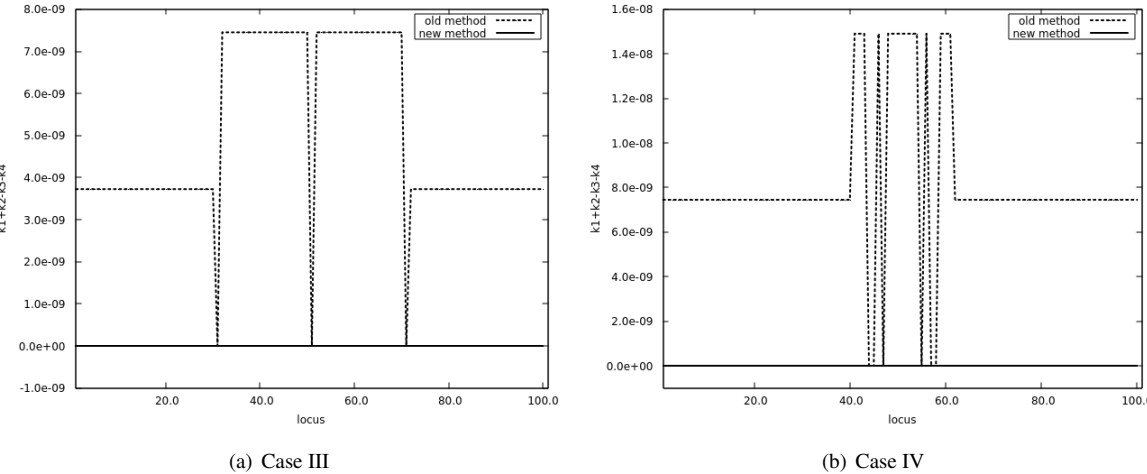

(a) Case III    (b) Case IV

**Figure 8.** The $\mathbf{k}_1 + \mathbf{k}_2 - \mathbf{k}_3 - \mathbf{k}_4$ variation from Case III (a) and IV (b). The iteration method (labeled 'old') produces small a residue for the sum of $\mathbf{k}_n$, but the explicit method (labeled 'new') gives no residue. Only sum of vector components is presented here.

## 5    Conclusions

It is shown that both methods can determine the resonance conditions, $\mathbf{k}_1 + \mathbf{k}_2 - \mathbf{k}_3 - \mathbf{k}_4 = 0$ and $\omega_1 + \omega_2 - \omega_3 - \omega_4 = 0$. However, the new method gives less residue. The residue is a number (its magnitude) of adding all solved wave numbers, this number should be zero. Therefore, the smaller the residue the better the method. The new method also omits iteration so it





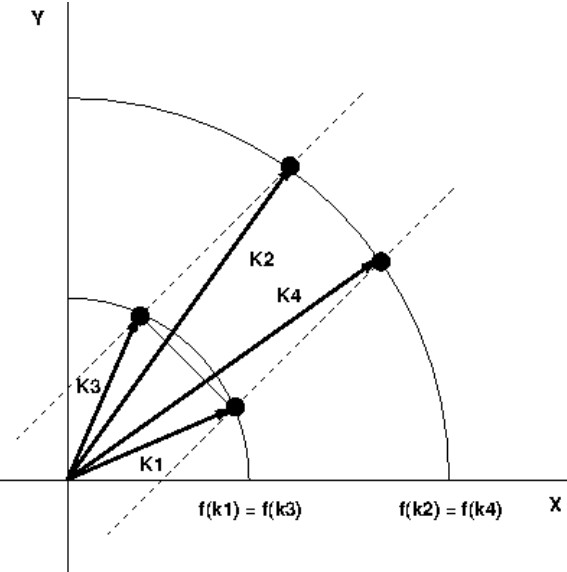

**Figure 9.** This is an illustration to depict that if $|\mathbf{k}_1| = |\mathbf{k}_3|$ then $\omega_1 = \omega_3$, also if $|\mathbf{k}_2| = |\mathbf{k}_4|$ then $\omega_2 = \omega_4$. The sum of $\omega_1 + \omega_2 - \omega_3 - \omega_4$ is zero when $\mathbf{k}_1 + \mathbf{k}_2 - \mathbf{k}_3 - \mathbf{k}_4 = 0$.

**Table 1.** Example of the resonance of $\omega$ in the deep water case.

|  | $k_x$ | $k_y$ | $k = |\mathbf{k}|$ |  | $\omega = \sqrt{gk}$ |
|---|---|---|---|---|---|
| $\mathbf{k}_1$ | 0.1000 | 0. | 0.1000 | $\omega_1$ | 0.9900 |
| $\mathbf{k}_3$ | 0.1575 | 0. | 0.1575 | $\omega_3$ | 1.2424 |
| $\mathbf{k}_2$ | 0.0420 | 0. | 0.0420 | $\omega_2$ | 0.6418 |
| $\mathbf{k}_4$ | -0.0155 | 0. | 0.0155 | $\omega_4$ | 0.3894 |
| $\mathbf{k}_1 + \mathbf{k}_2 - \mathbf{k}_3 - \mathbf{k}_4$ | 0.0000 | 0. | 0.0000 | $\omega_1 + \omega_2 - \omega_3 - \omega_4$ | 0.0000 |

can be concluded that the new algorithm works faster than the old one because no iteration required. Without iteration it also means a fixed grid is not necessary. This is important.

**Table 2.** Example of the resonance of $\omega$ in the shallow water case.

|  | $k_x$ | $k_y$ | $k = |\mathbf{k}|$ |  | $\omega = \sqrt{gk \tanh(kd)}$ |
|---|---|---|---|---|---|
| $\mathbf{k}_1$ | 0.1400 | 0. | 0.1400 | $\omega_1$ | 1.1021 |
| $\mathbf{k}_3$ | 0.2016 | 0. | 0.2016 | $\omega_3$ | 1.3808 |
| $\mathbf{k}_2$ | 0.1944 | 0.0321 | 0.1970 | $\omega_2$ | 1.3627 |
| $\mathbf{k}_4$ | 0.1328 | 0.0321 | 0.1366 | $\omega_4$ | 1.0840 |
| $\mathbf{k}_1 + \mathbf{k}_2 - \mathbf{k}_3 - \mathbf{k}_4$ | 0.0000 | 0. | 0.0000 | $\omega_1 + \omega_2 - \omega_3 - \omega_4$ | 0.0000 |





On the other hand, the explicit method can simplify the computation. For example in the Case I, to determine the distance between locus, $ds$, the old method will wait for all loci, $locus_1, locus_2, \ldots$, to be first found. Then it will do subtractions, $ds_1 = locus_2 - locus_1$, $ds_2 = locus_3 - locus_2$, etc. However, the new method can determine the $ds$ at the beginning without waiting until all loci computed. For example, because we know that in Case I, the loci is a straight line, then the $ds_i$ = length

of the line / (number of locus - 1), and $ds$ is uniform.

The existing wave model, for example SWAN or WAVEWATCH III$^{TM}$, computes the $S_{NL}$ based on the deep water (Case III) formulation. Applications where water is shallow of finite depth need to check again how the computation of $S_{NL}$ is derived. In this paper, we show 4 cases to determine the loci or domain of integration. Four subroutines to compute the loci can be easily built to handle any condition (deep or shallow water). Next, after we get the loci correctly, we have to examine other

variables in Eq. 1. Are the subprograms derived from formulations that suit the condition? However, actually finding the other variables is beyond this paper. This paper only shows how we start to do the computation for $S_{NL}$ using full Boltzmann integral correctly, by finding the loci based on the water depth. A correct procedure that is followed at the beginning of simulation will produce good results.

If we have proper subroutines that suit any condition of the real sea, then we can do simulation for hindcasts or forecasts

with better results. The applications to societal concerns and problems are numerous. Good simulations and forecasts of ocean wave data can make good contributions to studies related to ocean wave energy resources utilization, disaster prevention and reduction, and search and rescue, etc (Kamranzad et al. (2013); Mirzaei et al. (2013); Zheng et al. (2014); and Zheng & Li (2015)).

## 6    Code availability

The code written in FORTRAN is available on the following link
https://doi.org/10.5281/zenodo.1040466

*Acknowledgements.* The authors would like express their gratitude toward the Office of Naval Research, NERACOOS, MEOPAR, and PERD for supporting this project.





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
