# Peer review of "On Quadruplet Interactions for Ocean Surface Waves"

_Geoscientific Model Development, 2017_

## Short Comment (SC1) · 16 Nov 2017

The authors propose an improved technique for computing the rather complex integral expression describing the nonlinear energy transfer in an ocean wave spectrum. The integral transfer expression was derived and first computed by myself (J.Fluid Mech, 12, 481-500, 1962, and JFM, 15, 385-398, 1963, respectively). (The integral expression itself was first published in 1960, in Schiffstechnik, as yet without computations.). More detailed computations were later performed in the context of the JONSWAP wave growth field experiments (Hasselmann et al, 1973, Deutsche Hydrographische Zeitschrift, A8, 12, 1973).

The authors, however, give no reference to the origin and original computations of

the nonlinear energy transfer: their earliest citation is of a paper by Webb (Deep Sea Reseach, 1978).

I should, of cause, ignore my vanity and simply comment on the paper regarding its new results. However, here I must confess that I am no longer motivated today to look again into the complexities of the higher dimensional nonlinear energy transfer integration and the removal of the associated delta-functions. This is basically straightforward mathematically but algebraically tiresome. On the other hand, it is indeed a necessary task, since the nonlinear transfer has bee shown to be the dominant process governing the form of the windsea spectrum. With the enormous advances in computer power since our first exact computations in 1962 I have always been surprised that the later Direct Interaction approximations we introduced for operational wave models in 1988 (this is at least cited,J.Phys.Ocean. 18, 1988) have not yet been replaced by more exact numerical integrations.

Glancing through the paper diagonally I tend to agree with an earlier reviewer: the proposed exact numerical integration needs to be tested for its operational viability through an actual application in an operational wave model, for example at the European Centre for Medium Range Weather Forecasts. If this is carried out, and a more balanced history of the problem is presented (and ignoring my subjective feeling that the authors are making the numerics appear still more complicated than they actually are) I recommend publication.

---

## Referee Comment (RC1) · Anonymous Referee #1 · 26 Dec 2017

The authors describe a method for solving an equation used to determine the integration space for computing the non-linear four-wave interactions between wind-generated surface waves. The motivation of this research is highly flawed as the authors suggest the existence of a problem, which has already been solved satisfactorily more than 10 years ago. So, the manuscript offers a so-called solution for a non-existing problem. The only new element in this research is a mathematical detail avoiding iteration, but still offering a non-exact solution. Further, the manuscript contains some misquotations and various wrong statements and even a wrong figure. Based on these general remarks and some further issues detailed below I recommend a reject.

The manuscript describes a method for determining the position of the loci for k2 and k4 appearing in the WRT method for computing these interactions. For deep water,

explicit relations already exist, so this work does no add anything new. For shallow water, robust iterative methods exist, despite unbacked claims by the authors about poor convergence.

The work is not reproducible as for almost all figures no details are given about the chosen variables like k1, k3, step size, and depth.

Page #1. The title is incorrect. Not one interaction is solved. The manuscript only describes a mathematical detail offering a different method. Any consequence of their work on quadruplet interaction for ocean surface waves is missing. There is only some wild speculation that their method is important for wind-wave modelling and its applications.

**2/6: excessively large is a wild exaggeration. No reference or quantification is given. Further, the authors completely miss the fact that finding the integration space is only part of a pre-processing, whose time vanishes in comparison with applying the full integration method.**

**2/22: No criterion is given for better performance.**

**3/12: stating that the downshifting of the peak requires re-computation of the integration space is not needed. The authors define a non-existing problem. But, if there is need for such a re-computation, than proper references should be added.**

**4: For deep water, exact relations exist for determining the position of the loci. So, this work does not add anything new.**

**5, Fig 1: Details are missing of k1, k3 and depth. So this is not reproducible**

**8/Figure 2: it is impossible that the shallow water egg-wise locus extends outside the circle through rmin and rmax.**

**10/10: The authors claim to have an accurate result without iteration. This is still not an exact answer and no criterion is given. It is still an approximation.**

**12, No details are given about step sizes in creating a the points on the locus.**

**13: In Resio and Perrie (2008) no mention is made of any method determining the loci, let alone the number of iterations needed. The number 200 is a wild statement without any background information on the method applied or accuracy criterion applied. This misquotation is a severe flaw of this manuscript.**

**14: the results in the figures 7 and 8 are anecdotal. It is just one example and possibly out of context. It is also a matter of choosing a scale to magnify any difference without a quantitative context.**

**14/3. No reason is given why the residue should be zero. In numerical modelling, and considering the whole chain of steps needed to compute the full Boltzmann integral, errors appear at various levels, and it requires much further research to determine any effect on the total result, whatever that might be.**

**15 The phrase 'this is important' is not substantiated.**

**16/6: The statement that the Snl method implemented in Wavewatch III and SWAN only considers deep water is wrong.**

**16: the last 5 lines are true in itself, but these have no relation with the work presented in this manuscript.**

---

## Referee Comment (RC2) · Anonymous Referee #2 · 8 Jan 2018

The manuscript addresses some details of computing nonlinear four-wave interactions for ocean surface (gravity) waves. In fact, the paper only addresses details of the structural reduction of the multi-dimensional integration space, and as such represents only a detail of a detail of the technical issues occurring with calculating these interactions, making it a niche paper indeed. With that, one could argue that this should be a technical report, rather than a scientific paper, not in the last place, as this manuscript seems a little incomplete with respect to the results. Having said that, the manuscript is well written and easy to follow, and its results may be useful for the small group of researchers working on the full Boltzmann integral version of these interactions. Below are some more detailed remarks on the manuscript

1. I agree with Klaus Hasselmann in his comments on this paper that the references

in the introduction are insufficient. Only using the Webb (1978) reference does not give due credit to Klaus (and Owen Phillips and Vladimir Zakharov) regarding their fundamental contributions to this field. And a reference only to the authors' two-scale approach does not give due credit to the plethora of newer Snl approaches, some of which have even found their way into operations at NOAA. This does not influence the core of the paper, but it represents unnecessary discourteousness.

2. At its core, the present manuscript seeks to remove the need for iterations in solving the dispersion relation as needed for determining integration loci. I wonder if this could be achieved even easier by using the non-dimensional interpolation table used inside of WAVEWATCH III. Have the authors considered this approach, and how would that approach compare to the new method both in cost and accuracy? Looking at the code, the cost seems much smaller, and the relative accuracy should be assessed without much effort, and the code used in WAVEWATCH III is written as a self-sufficient package than can be used in any code.

3. The manuscript is incomplete. The new method is advertised as more accurate and cheaper. The accuracy is quantified, but the cost is not. The cost aspect is likely more convincing for its use by others. How much cheaper would the entire exact interaction be with the new approach? Considering the experience of the authors with the Resio codes for exact interactions, it seems a small effort to actually compare the cost of this Snl package with the old and new approaches, both for calculating interactions for a given spectrum, or for full model integration with all source terms.

4. OK, this last observation may well be taken as being a little over the top. Snl in its exact form is very sensitive to details of computations. I would love to see some proof that the interactions obtained with the new approach indeed are realistic without unexpected features or noise, and that they result in stable model integration. I expect that changing these technical details will have little impact on the resulting interactions or model integrations, but past experience has taught me to be overly suspicious in this respect.

---

## Author Comment (AC1) · 9 Jan 2018

Dear Reviewer,

Thank you for all comments. These comments will be used to improve our paper in the future and we'll put your name in references. However, considering our scope of the topic and demand of GMD readers, we would like to withdraw the paper from GMD. Thanks again for the inputs. We apologize for any inconveniences.

Kind regards A Susilo

---

## Author Comment (AC2) · 9 Jan 2018

Dear Reviewer,

Thank you for all comments. These comments will be used to improve our paper in the future. However, considering our scope of the topic and demand of GMD readers, we would like to withdraw the paper from GMD. Thanks again for the inputs. We apologize for any inconveniences.

Kind regards A Susilo
* * *